# Leadership models in era of new technological challenges in construction projects

Jarosław Górecki[1]*, Ewa Bojarowicz[1], Jadwiga Bizon-Górecka[2], Umer Zaman[3], Abdullah Emre Keleş[4]

1 Bydgoszcz University of Science and Technology, Bydgoszcz, Poland, 2 University of Bydgoszcz, Bydgoszcz, Poland, 3 Endicott College of International Studies, Woosong University, Daejeon, South Korea, 4 Department of Civil Engineering, Alparslan Türkeş Science and Technology University, Adana, Turkey

⊛ These authors contributed equally to this work.
* gorecki@pbs.edu.pl

**Data Availability Statement:** All relevant data are within the paper and its Supporting Information files.

## Abstract

The building sector is under the significant influence of emerging technologies. Structures shape the environment and "consume" natural resources throughout their life cycle. They "live" many years after the construction which implies a dependence on some generations of supporting technologies. They can be useful in the subsequent phases: design, construction, maintenance and demolition. They may refer to main processes (construction production) as well as to concurrent processes (management, accountancy, logistics etc.). Computers, automated tools and machines or other intelligent devices seem to be inevitable in the 21$^{st}$ century. Therefore, contractors of construction projects should be sensitive to these issues. Based on literature studies, the article revealed that knowledge management in a construction company should primarily rely on the corporate culture that manifests a preference for computer-aided methods. This part was supplemented by a questionnaire technique and a statistical analysis of the results. It was concluded that the path to technological maturity of the construction company is a continual process. Consistency in this pursuit enables effective promotion of innovative technologies in the construction company. The research allowed us to draw three explicit phases: lack of experience, euphoria, and experience in becoming a technologically matured enterprise.

## Introduction

Construction as a significant sector of the economy has been a subject of the continuous impact of technology. After the Second World War, the development of the technology was concentrated on the methods of constructing prefabricated buildings and non-building structures. A straight advantage of the approach was a rapid construction process which resulted in a quick supply of the ecosystem needed by and convenient for societies in many European countries after years of destructive military activities. After some decades, technology development was focused on other issues, e.g. manufacturability [1], energy savings [2], quality [3],

**Funding:** This research received funding from the Bydgoszcz University of Science and Technology, Poland (Grant no. BN-WBAIS-0/2022).

**Competing interests:** The authors have declared that no competing interests exist.

cost and time efficiency [4, 5], diversification of solutions etc.–it means a further improvement of the construction methods. Nowadays, development in this area is still observed (e.g. graphic or transparent concrete [6], new and improved precast accessories, including seismic-resistant joinery techniques [7] etc.), but the effects are no longer as spectacular. On the other hand, brand new materials are expected to replace the currently used, and not always environmentally friendly material technologies.

A final phase of the 20th century related to another technological revolution–the implementation of IT in construction projects. It was noticed that methods of construction were developed sufficiently but there were still other undiscovered challenges connected with an efficient way of raising buildings and creating infrastructure. Digitalization of the construction sector is not a rapid process. Unlike in other sectors of the economy (manufacturing or services), market players of the construction are still quite sceptical about the innovations of the IT origin. Many construction companies are featured by a comparably low level of use of computer-aided methods. Due to this fact, there is still a small percentage of enterprises using Building Information Modelling (BIM) [8] or accompanying technologies Virtual Reality (VR) [9], Augmented Reality (AR) [10] or Mixed Reality (MR).

Lately, it has been discussed that sustainable ways of conducting construction processes are inevitable. Therefore, more and more theoreticians and professionals concentrate on new concepts like environment-focused certifications of buildings including the Building Research Establishment Environmental Assessment Method (BREEAM) [11] or the Leadership in Energy and Environmental Design (LEED) [12], the ISO 14000 and its relevance to the construction industry, the Life Cycle Assessment (LCA) [13, 14], Life Cycle Costing Analysis (LCCA) [15], cradle-to-cradle (C2C) approach [16, 17], design for deconstruction (DfD) [18, 19], green labels [20] or Circular Economy (CE) [21–23]. All these ideas seem quite complicated, but they all have something in common—they relate to mechanisms that are favourable, not harmful to the environment, and related to construction projects. Despite the apparent environmental connotation, these activities are possible mainly due to the use of product or process innovations—whether in the field of heating, ventilation, and air conditioning (HVAC) technologies, whether in the field of new recovery, reuse and recycling technologies, or finally due to the use of new, yet even uncovered or untested building materials.

The development of construction-related technology has not been limited by the aforementioned actions. The Industry 4.0 (or its related term of Construction 4.0) [24, 25] concept brings even more challenges. It requires changes on many levels: technological, organizational, and mental both in construction companies and construction projects. It seems that despite the unimaginable development of the entire construction ecosystem including all elements of business activity in construction, the upcoming years may be a moment of a critical juncture in which it will be possible to combine seemingly disjoint and incoherent elements into one idea of sustainable and circular construction. Remodelling the entire traditional construction process into a chain of interrelated activities carried out by automated production systems (like 3D printing [26]) based on Big Data [27–29], processed in data mining supervised by artificial intelligence (AI) [30, 31], following the spirit of sustainable development, with a closed-loop of resources conditioned by product and process innovations is becoming the industry's most important challenge in 21st century.

All in all, the subject of trends in the construction sector should be broadly perceived. This article focuses on the issue of technological conditions for the development of construction companies. It has been assumed that the technological process is one of many processes present in the company's production system (Fig 1).

However, it requires a constant technical support. Preferring a development model based on technology treated seriously as an advantage and an implementation of innovative

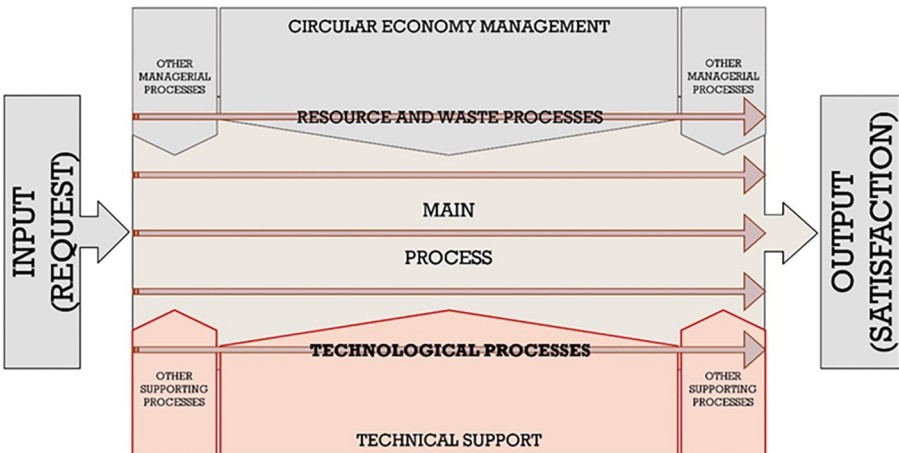

**Fig 1. Business process map in companies involved in construction projects–specification of supporting processes including technical support.**

technologies as an opportunity both for companies and projects has become the starting point of the study.

Due to the dynamic development of construction companies and the fast implementation time of investments (short construction phase), as well as increasingly higher standards of execution of facilities, the problem of leadership is often discussed, which largely affects the success or failure of investments. Research in this direction has been carried out by thousands of scientists and is still being carried out. Each result of the conducted research reveals new aspects of leadership; hence the studies should be continued, supplemented with new conclusions, additional interpretations, models, or new theories. The constant interest in the topic of leadership results from the specificity of their conduct—they are performed based on people's behaviour; therefore, each subsequent research may provide a new perspective on the topic in question. This article is a response to this gap. The main aim of this article is to present the prevailing management styles at construction sites. As a result of the refinement of the scope of the research, several specific objectives were focused on: analysis of the survey results in terms of potential discrepancy in responses in individual groups of respondents and building a model of changes in leadership patterns in the context of the managerial behaviour mix. The work consists of three main chapters, an introduction, a discussion of research results and a section of conclusions. It was assumed that there is no one, dominant model, and supposedly a mix of styles that should be matched to a given situation is the most popular. As a result of the study, an evolutionary leadership model for effective knowledge diffusion about technology in a construction company was created.

## Theoretical background

For civil engineers, to build simply means to erect buildings and other type of infrastructure. From the executive point of view, the erection process can be categorized according to where most construction production takes place. Therefore, it is possible to build in delivered materials directly on the construction site (e.g. monolithic construction). Another option is prefabrication (pre-construction) of elements that, after being transported to the construction site, are in-built to the right place, i.e. jointed using various joinery techniques [32]. The further division consists in considering the dominant material used to manufacture a given building element (concrete—precast, reinforced concrete, ceramic, wooden, metal structures—

although the latter are only prefabricated). Today, it is believed that it is easier to implement innovations regarding the quality criterion in a prefabrication mode [33, 34].

Mechanization of the production of building materials is a part of the economy that drives the development of technology basically around the construction sector, but which has a significant impact on this sector. It turns out that innovations derived from manufacturing materials create pressure to seek new solutions at the stage of construction works. Prefabrication factories are a subject of the robotization of production processes. To meet the expectations of the customers and not to squander a high quality obtained at an earlier stage, contractors are somewhat forced to demonstrate continuous improvement of processes at the further step–on a construction site. What is more, innovations from the design stage also reinforce the need for implementing innovations on the construction site. The best example of this is the use of BIM. Software, based on such an attitude, is a great tool for simulating the best solutions and creating the so-called Digital Twin [35, 36] of a future facility. However, the potential of BIM is unused if technologies based on this idea are not used in subsequent life cycle stages of a building or a non-building structure, i.e. at the stage of construction, maintenance/operation or end-of-life. All in all, the conceptual package related to the development of technologies used in construction is very capacious and contains a lot of concepts. A set of these ideas are presented in the form of a word cloud in Fig 2.

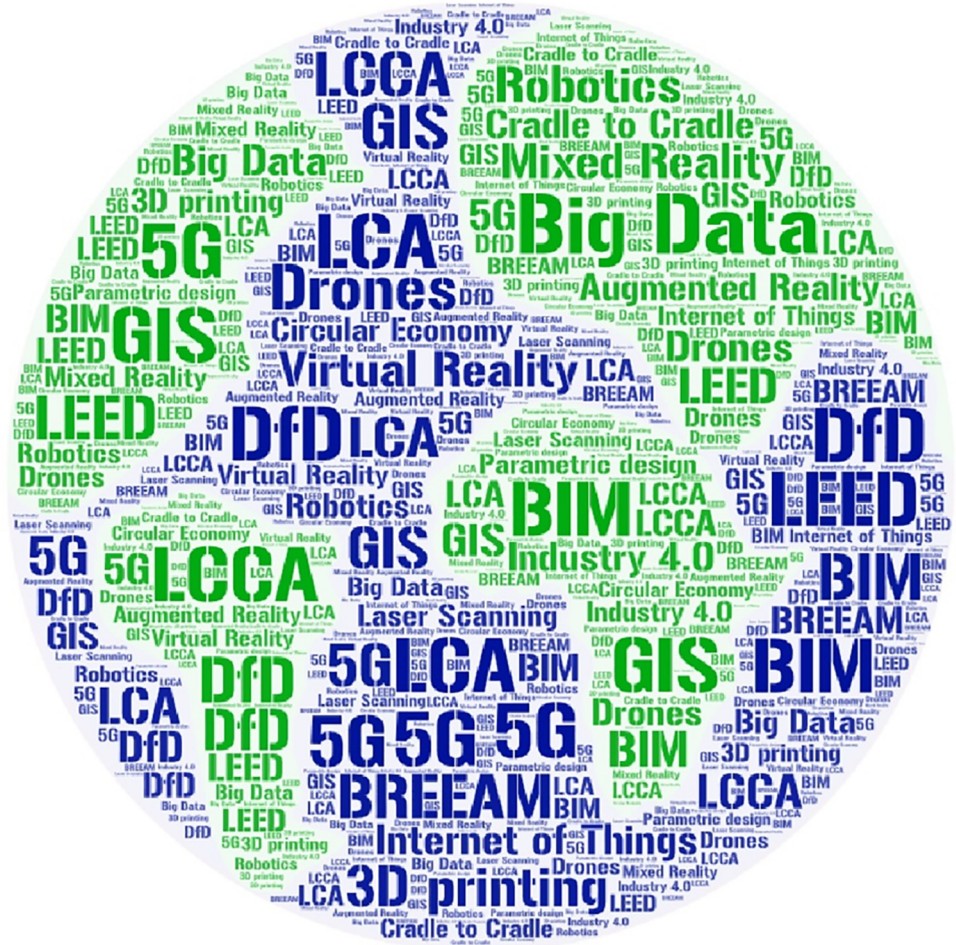

**Fig 2. Word cloud based on concepts connected with new technologies connected with the construction sector.**

The development of innovations can be based on the following three pillars: technology, law and administration as well as social acceptance.

Historically, technology has always been the driving force of the industry. Today, thanks to the universal availability of technology, as well as the remodelling business concepts of various industries, it is a competitive advantage mainly in the sphere of services.

It is worth noting that the construction technology is undergoing gradual changes. However, nothing influenced its development more than the digital revolution [37–39]. It has become a catalyst for the development of enterprises, improvement of risk management, sustainability etc. The advancement of digital technologies made them popular, and at the same time led to significant cost reductions [40]. Today it is difficult to imagine a serious construction project without the use of a computer, sending e-mails or telephone calls. These seemingly obvious processes save stakeholders' time and money. The abandonment of printing documents and the promotion of digital construction also brings relief to the natural environment and makes construction activities more sustainable.

Reliable communication is of great importance for improving the business profitability of any kind [41, 42]. Nowadays, more and more often communication is carried out at a distance by virtual communication channels [43]. For at least 20 years, mobile phones have become the most popular technical means. Fixed broadband is also becoming more and more significant. The latter encompasses any high-speed data transmission to a fixed location (a house or an office) based on a variety of technologies, including cable, fiber optics, and wireless connection. Its advantage is high-speed Internet.

According to reliable statistics (e.g. The World Bank's), access to communication technologies, and hence their popularity, are diversified. In rich and developed parts of the globe (e.g. Euro area), the number of mobile cellular subscriptions has stabilized at around 120/100 people, whereas in Africa this level is estimated at 60–80 per 100 people (Fig 3).

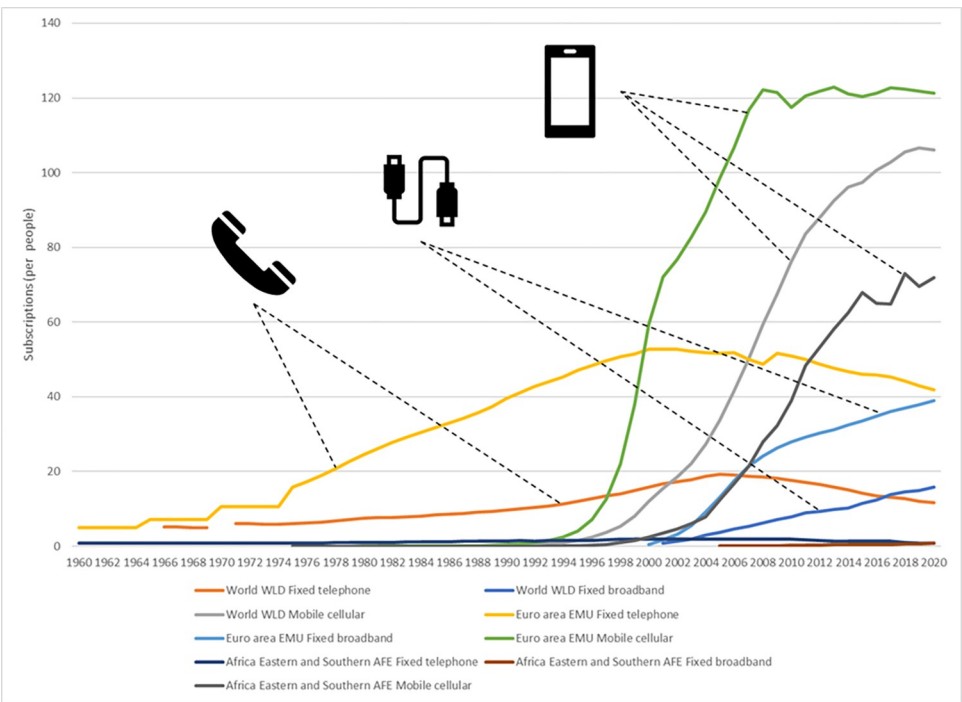

**Fig 3. Popularity of communication technologies over last 60 years according to [44].**

Such irregularity, first of all, may mean that the management of construction projects in less "digitized" areas will require traditional methods of communication. On the other hand, a construction project in which the circulation of documentation (often of size of several gigabytes) is electronic, requires an access to high-speed Internet. This condition limits the range of digital construction to those areas that are already economically developed.

So, the first pillar of innovations is easily achievable thanks to advanced computational techniques or modern production systems that can provide a product whose specification is unimaginably complicated. There are some factors determining the use of the appropriate technology, i.a. promoting factors (expectancies, task values) or actively detracting values (mainly costs) [45]. Other industries provide the construction industry with technologies, and their fortune depends on different aspects, including economic conditions [46]. Intriguingly, will to invest in digital technologies leads to organisational adoption failures [47]. Technological innovations that appear from time to time are usually an opportunity to further improvement of processes. Therefore, in construction companies special posts are sometimes set up to monitor and manage such novelties when they become affordable [48].

However, the second pillar called "law and administration" seems to be a little bit harder to manage. Legal regulations and procedures take place at various levels and forms. Governments and international institutions introduce them to prevent uncontrolled processes. Supporters of deregulation and liberalization of the economy are opposed to far-reaching changes in this area. On the other hand, some restrictions, e.g. those relating to obligatory rules of conduct, aimed at protecting the environment, work and data safety, or the protection of human life and health, are deeply justified. These regulations also often apply to the sphere of innovation, e.g. blockchain-based cyber systems [49].

Innovations in construction include also new material solutions. However, they require time-consuming and long-term certification and admission procedures on the market. For instance, in the European Union the Construction Products Regulation (CPR) lays down harmonised rules for the marketing of construction products in the EU. The Declaration of Performance is a key part of that [50]. Each construction product needs this Declaration and has to be CE marked. It is mandatory for certain product groups to be sold within the European Union, the European Free Trade Association (EFTA), and Turkey. The procedures and legal consequences connected with counterfeiting of the CE marking vary according to the respective member state's national administrative and penal legislation. Outside the European Union, similar solutions sometimes exist. For example, Great Britain adopted the UK Conformity Assessed (UKCA) marking according to the Brexit policy.

In addition to product innovations, there are also process, organizational and marketing innovations. The reorganization of processes related to a construction project requires the use of modern technological solutions, and these have recently been most often associated with BIM. It allows all stakeholders to have access to the same information, at the same time, through the interoperability of technology platforms. So new challenges arise like cybersecurity and respect for digital ownership or intellectual property [51, 52]. A protection against unauthorized online access mean not only anti-theft issues but also to ensure digital security [52, 53]. Data leakage can sometimes contribute to significant sabotage attempts what, in turn, may result in a reduction in users' trust to digital technologies. Thus, construction projects appear as a discipline where both safety and security become key success factors [54, 55].

The aforementioned decision-making methods based on detailed analysis and data mining, especially thanks to the advanced information technology enabling the use of Big Data, are becoming more and more popular. Collecting data about people or their property is also subject to legal regulations. Many countries have introduced specific restrictions on public privacy. For instance, Germany was the first European Union member state to adopt a national

law implementing the General Data Protection Regulation (Regulation (EU) 2016/679) ('GDPR') in the form of the Federal Data Protection Act of 30 June 2017 ('BDSG') [56].

However, the most difficult challenge is to convince users and project stakeholders to accept, adopt and use innovations. People and their perceptions are becoming a very crucial factor for the expansion of innovations. A social acceptance (the third pillar) seems to play a significant role in a diffusion of the new methods of act. The main barrier is primarily people's natural resistance to change. Conformity often means that the improved solutions are adopted with a certain delay. However, there are many factors accelerating adaptation to change. The conviction to innovate can be achieved in many ways, but the most important motivator is the awareness of profits in the form of lower costs, shorter service time, better quality, higher level of satisfaction, etc. Moreover, some investigations show that a guidance during the implementation of the innovation on the construction site increases the likeliness of acceptance of the digital innovation on-site [57].

Stimulating the awareness of users can be effective in the case of customised (tailored to the needs of a specific person) messages. They can be communicated through applications that filter people's needs and expectations and choose the most effective set of advice based on their preferences. A good example would be here Sustainable lifestyle test (https://sitoumus2050.fi/en/web/sitoumus2050/lifestyles#/). Such tools can help to break down barriers and raise people's awareness [58].

Therefore we are quite sure that technology, as well as the progressive digitization of life, can play a significant role in changing the behaviour of stakeholders [58, 59]. At the same time, individual access to information technologies and the Internet services are indispensable, although this occurs with various results in different parts of the world (Fig 4).

Construction works the same way as it did many years ago. The conditions in which construction projects are carried out remain unchanged—still, the building must ultimately be erected (whether constructed or assembled) in the target area, in certain variable soil and

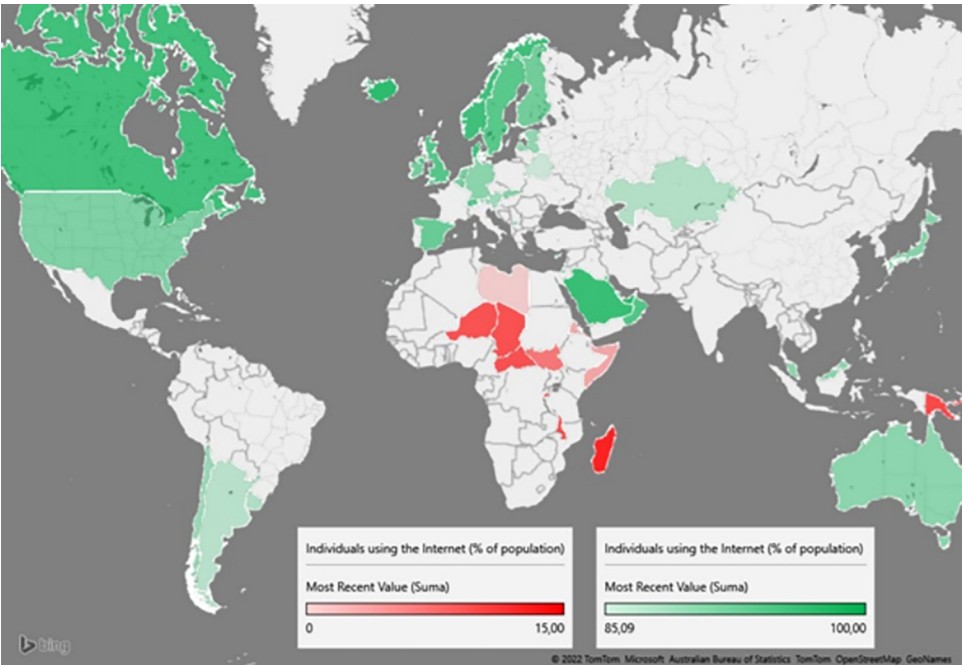

**Fig 4. Discrepancy of the Internet connectivity in various parts of the globe.**

water conditions, with the participation of many entities, not always agreeing with each other, etc. However, the methods and means of achieving the same goal as before have been and still are being improved. To sum up, technological innovations in construction are present in various areas, from the production of building materials, design processes, logistics processes, methods of erecting construction objects, including technologies of construction works—junctures and constructing, techniques of project management (in costing, scheduling, quality control, inspections or project settlement), up to the stage of managing the existing building or non-building structure, and finally the moment of its demolition.

That is why innovations and technology play such an important role in the construction sector. The process of familiarizing people with new solutions is often long-term and time-consuming. It requires the involvement of both parties (the learner and the scholar). It must be embedded in the culture of the organization as an inherent feature of the knowledge management system.

## Materials and methods

A crucial factor for the proper functioning of any organization is the selection of the appropriate management staff, who will effectively influence the achieved results through leadership using efficient management. Therefore, the primary goal of this article is to examine the leading leadership models in construction projects (temporary organizations) and identify the model that leaders should pursue to achieve their goals, especially in the context of emerging technological challenges.

Efficient management is based on a multi-phase decision-making process that has an impact on economic, formal, legal, and technical development. Management is closely related to information processes. Decisions are made on their basis, therefore further development depends on their quantity and quality, which should anticipate possible inconveniences and complications related to the ongoing changes. Moreover, the information process is an important element in defining management, it has a significant impact on decisions made. Due to the dynamically developing environment, the approved and implemented choices may vary during their implementation—alternative decisions are made.

In each organization, including construction projects, there can be distinguished interactions between employees. The emotional links that are built between them significantly affect the work performed. They result from the position taken (the post) and the way of communication between different levels of management power. The majority of the links may affect the spontaneous choice of the type of organizational structure. On small construction sites, the organizational structure diagram is not complicated [60], especially in a situation where the site manager has limited management capabilities. In such a case, some decisions are transferred to the management levels. These functions are usually performed by the owners of companies. Comparing the organizational structure of a large construction site can lead to noting that the site manager has broader competencies and decisions are made mainly by him. Often this is due to the location of the headquarters. The company is located far away from the project location. The manager of this type of organization is independent and plays a similar role to the director on a given construction site. In this case, the structure of a large construction site can even be compared to the structure of the entire enterprise or at least a significant part of the company [61].

A leader should be known for her/his need to achieve goals. The person should be willing to exercise power of varied scale (power over/to/with) [62], be self-confident and have initiative and instil trust among group members [63]. In addition, he or she should have no difficulties with problem-solving, be creative, think rationally, and when necessary, show flexibility in

managing the entrusted tasks [64]. The player should demonstrate competence in creating a vision, which is crucial to introduce any changes. The leader must be able to anticipate the consequences of decisions made and have a good insight into all emerging challenges [65].

The leadership model should be adapted to the actual demand resulting from the current situation, e.g. in the case when the deadline of the construction project is approaching, the situation requires adopting the model of an autocrat manager who, by focusing on the task, will fulfil the facility's completion within the guaranteed time, regardless of the negative atmosphere in the team caused by time pressure or stress. The autocratic model based on tasks, not on mutual relations in the team, is in this case more favourable and gives a greater probability of success. On the other hand, e.g. in a situation where the investment is carried out at the correct pace—there are no delays, there is no pressure associated with time—the manager should adopt a democratic style that is focused more on people and mutual relations.

To discover an opinion about the major leadership models existing on construction sites, surveys are commonly used methods. These are research methods consisting in preparing questions and providing written answers. They are grouped as empirical methods. The issues included in a questionnaire should not be long and elaborated, they should be consistent and logical and not cause fatigue or weariness when completing the questionnaire. The questionnaire should be structured transparently.

In the first stage, questions are formulated for which it is intended to get answers. Then, some hypotheses that describe the relationships between the obtained results should be stated. In the questionnaire research, the scope of the research should be specified, as well as information to which community group or organization it is assigned.

The questionnaire research will allow for a quantitative and qualitative look at the research topic. Thanks to the results, it is possible to determine the causes of behaviours. The results can be presented in a descriptive manner or graphs, either as a percentage or a number.

After conducting and collecting the data, a questionnaire should be developed, and conclusions confirming or contradicting the hypothesis drawn.

The multiple leadership questionnaire (MLQ) scale developed by Bass and Avolio and used by one of the authors in his paper [66] was used as the basis for determining the questionnaire questions of this study [67]. The questionnaire defines the 3 leadership types discussed in this study and consists of 45 statements on a 5-point Likert scale. In the research, 9 of 45 questions, which were the most decisive, were selected and asked the participants during a session of the questionnaire. In this respect, it is clear that the answers to these questions can be used in determining the leadership types aimed at the study.

To verify which leadership model is leading on construction sites, a survey was conducted using a questionnaire which consisted of several issues listed in a traditional sequence: screening and rapport, product-specific and then demographic questions [68].

The questionnaire was sent to site managers and construction managers working on chosen construction sites in Poland. The questionnaire was sent by an electronic method (e-mail) from 15/01/2021 to 28/01/2021 (within 2 weeks) which was conditioned by limited access to other, especially personally administered, methods due to the COVID-19 pandemic. 41 questionnaires were received out of 60 sent out, which gives a response rate of 68.3%.

The screening phase included questions focused on the role of the respondents at the construction site and their experience. Among the surveyed sample everyone was experienced in the execution of construction projects and represented a leader's point of view which enabled a further step of the study.

Regarding product-specific questions, it was intended to ask nine scaled questions related to three specific leadership models (3 questions to each): transformational (TF), transactional (TA), and passive/avoidant (PA)–Table 1. The rating scale consists of five levels, from 0 to 4,

**Table 1. A list of questions asked in the survey.**

| No. | Question | Matched style |
|---|---|---|
| 1 | How often do you go beyond your self-interest or group welfare? (TF-1) | TF |
| 2 | How often do I have a sense of strength and self-confidence? (TF-2) | TF |
| 3 | How often do you let others see problems from different perspectives? (TF-3) | TF |
| 4 | How often do you clearly state what to expect when the business objectives are achieved? (TA-1) | TA |
| 5 | How often do you focus your attention on dealing with mistakes, complaints, and errors? (TA-2) | TA |
| 6 | How often do you track all the errors? (TA-3) | TA |
| 7 | How often do you stay out of the way until the problems get serious? (PA-1) | PA |
| 8 | How often do you avoid making decisions? (PA-2) | PA |
| 9 | How often do you delay answering urgent questions? (PA-3) | PA |

showing how often the respondent shows the given features. The frequency of occurrence of a given leadership style represented by a respondent (according to his/her opinion) can be 0 (if the context, described in the question, never takes place), 1 (if rarely), 2 (if sometimes), 3 (if quite often), and 4 (if very often/always).

Since an electronic method was used, all the responses were collected and saved in a database file. An analysis of data included a verification of completeness and consistency of the gathered data. The next part of the article presents main results of the survey.

## Results

Among the respondents, a minority (19.5%) was female which gives a little overrepresentation in comparison to statistics from the U.S. - 9.9% of employed people in the construction sector are women [69]. However, these statistics include all posts in the sector whereas if we skip a direct workforce from the total number of workers, the percentage of the civil engineering workforce composed of women arises up to 14% [70]. In this context, the divergence seems to be insignificant.

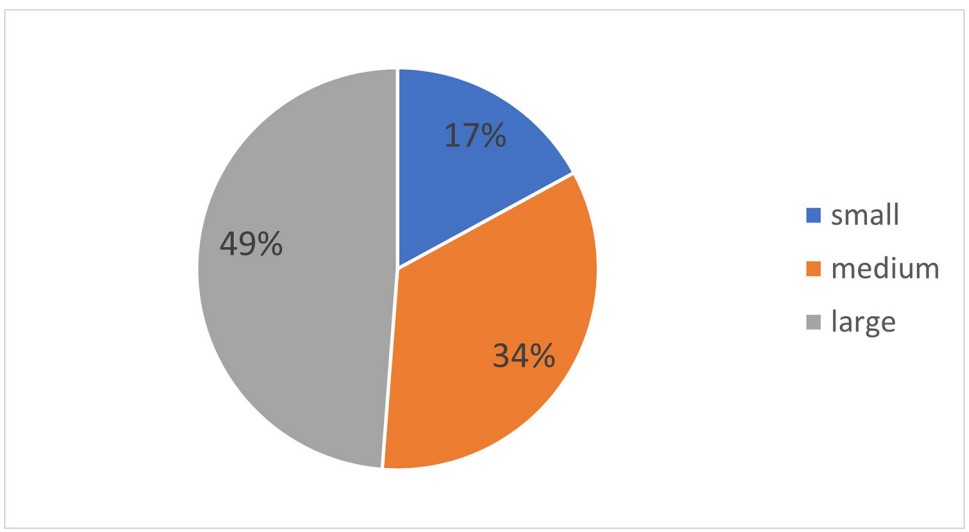

**Fig 5. Size of the enterprise represented by the respondents.**

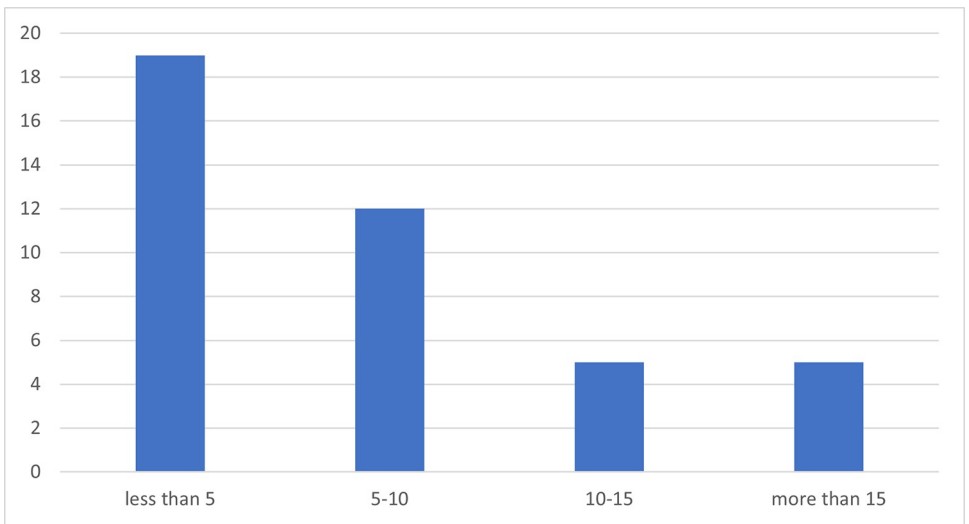

**Fig 6. Distribution of experience in managing construction works represented by the respondents.**

Among 41 respondents, the most numerous were employees of large companies (more than 250 employees). None of the respondents belonged to micro companies (less than 10 employees). Details of this feature are shown in Fig 5.

The largest group of respondents are people with less than 5 years of experience (around 47% of the total number of respondents). The least numerous group are people with experience longer than 15 years (see Fig 6).

Demographic questions defined the features of a leader model.

Figs 7–9 reflect the respondents' views on their managerial behaviour in relation to questions 1–3.

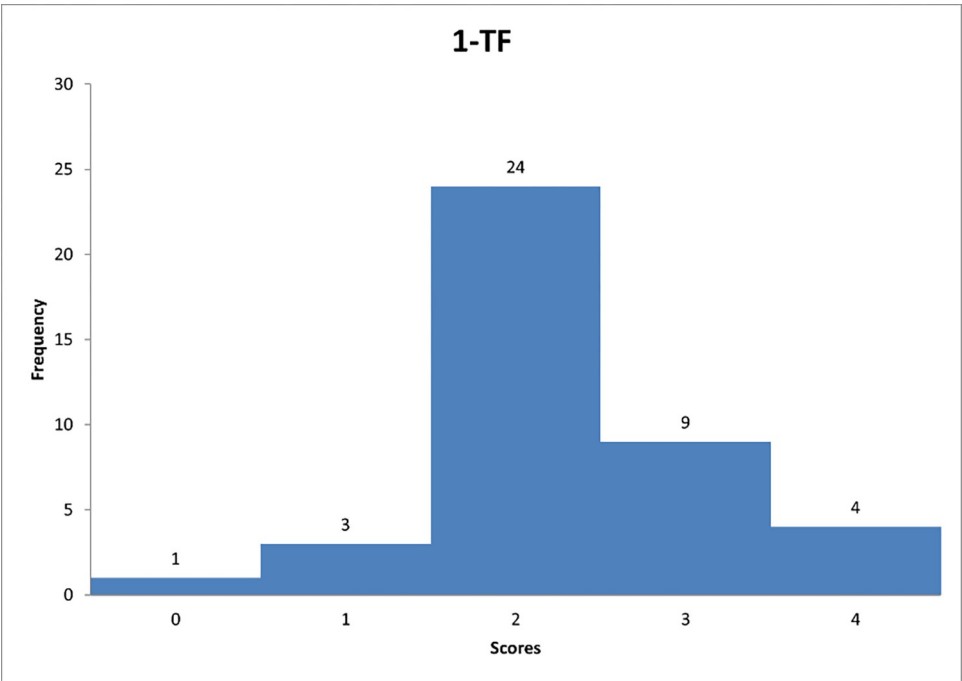

**Fig 7. Histogram based on respondents' propensity to adapt transformational style in relation to question TF-1.**

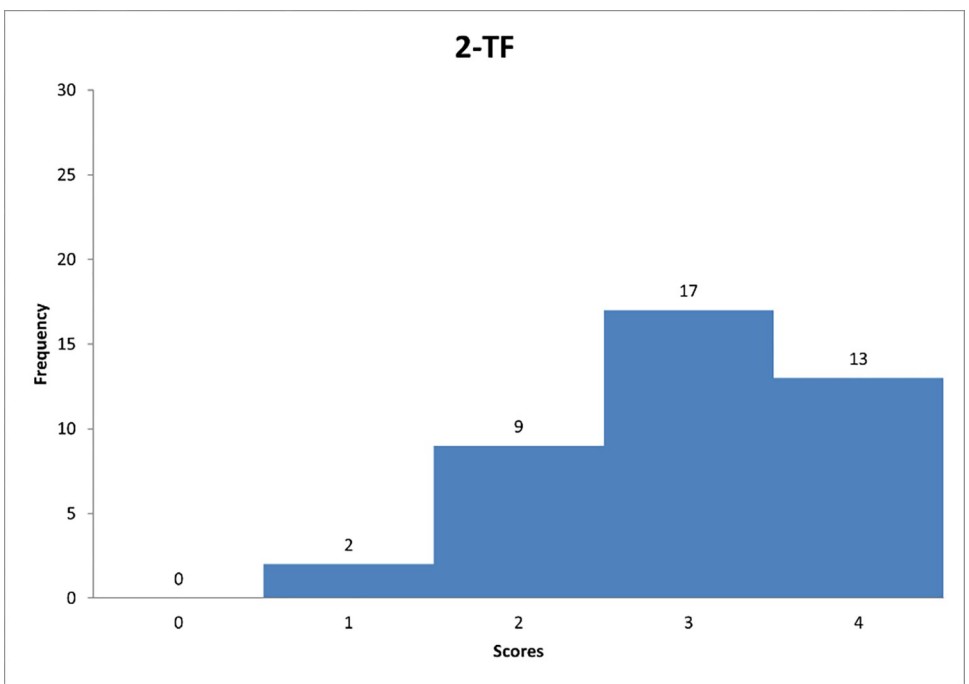

**Fig 8. Histogram based on respondents' propensity to adapt transformational style in relation to question TF-2.**

Figs 7–9 can support the features of a transformational leader model.

Figs 10–12 reflect the respondents' views on their managerial behaviour in relation to questions 4–6.

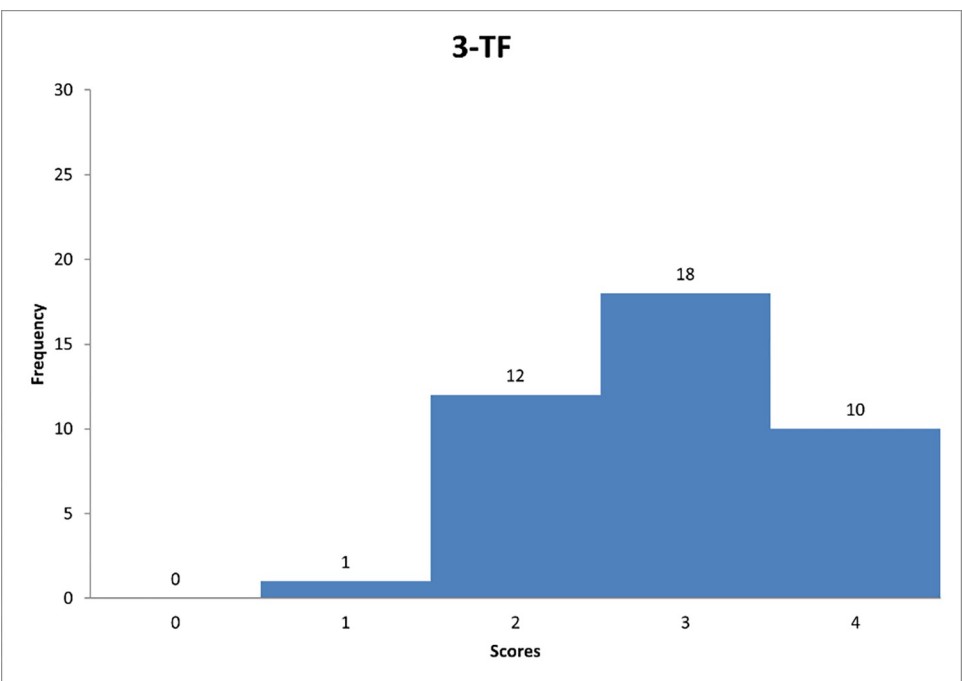

**Fig 9. Histogram based on respondents' propensity to adapt transformational style in relation to question TF-3.**

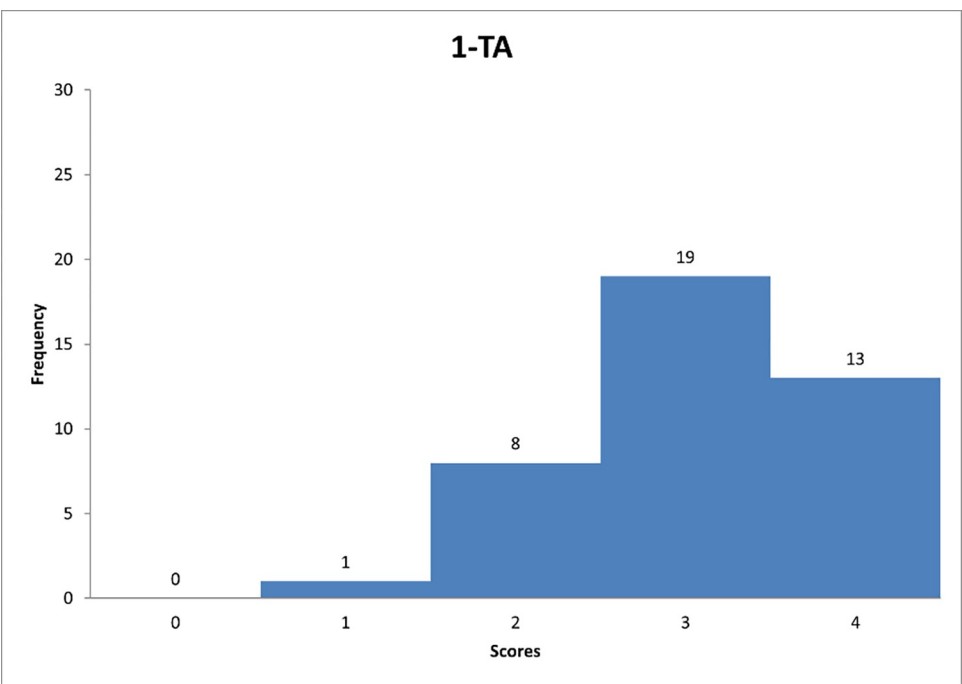

**Fig 10. Histogram based on respondents' propensity to adapt transactional style in relation to question TA-1.**

Figs 10–12 can support the features of a transactional leader model.

Figs 13–15 reflect the respondents' views on their managerial behaviour in relation to questions 7–9.

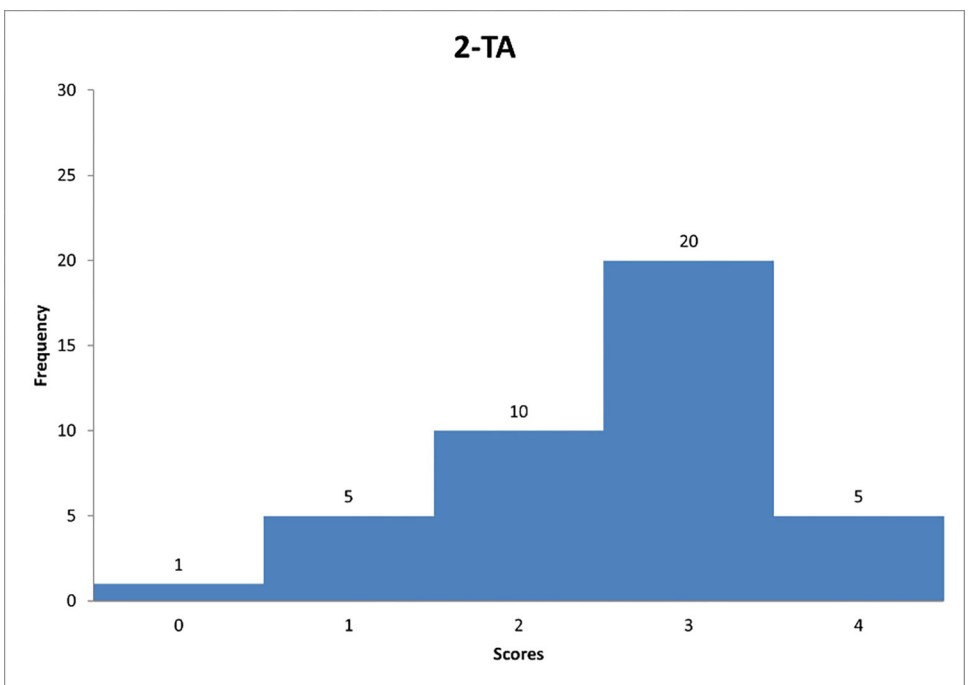

**Fig 11. Histogram based on respondents' propensity to adapt transactional style in relation to question TA-2.**

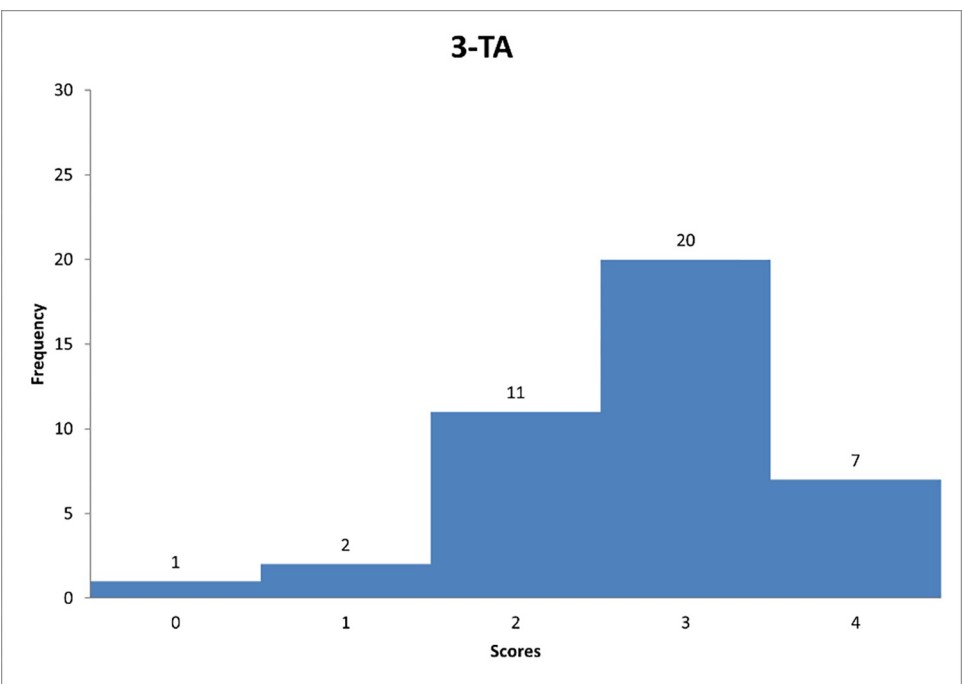

**Fig 12. Histogram based on respondents' propensity to adapt transactional style in relation to question TA-3.**

Figs 13–15 can support the features of a/an passive/avoidant leader model.

Detailed data analysis provides more information. For example, in Fig 7, it is worth noting that the largest number of respondents showed a tendency to go "sometimes" beyond self-

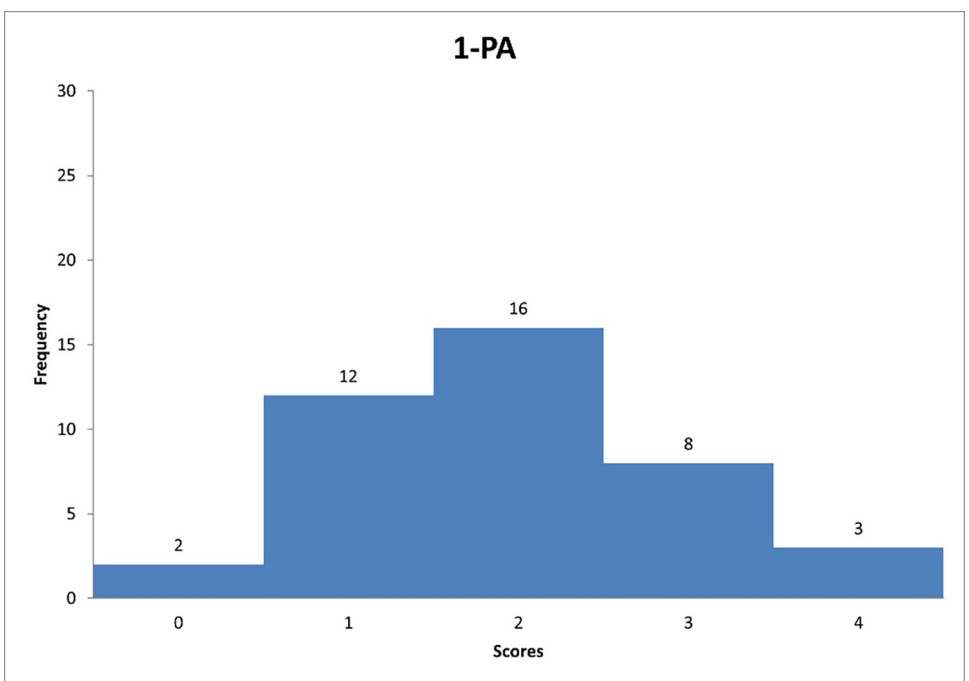

**Fig 13. Histograms based on respondents' propensity to adapt passive/avoidant style in relation to question PA-1.**

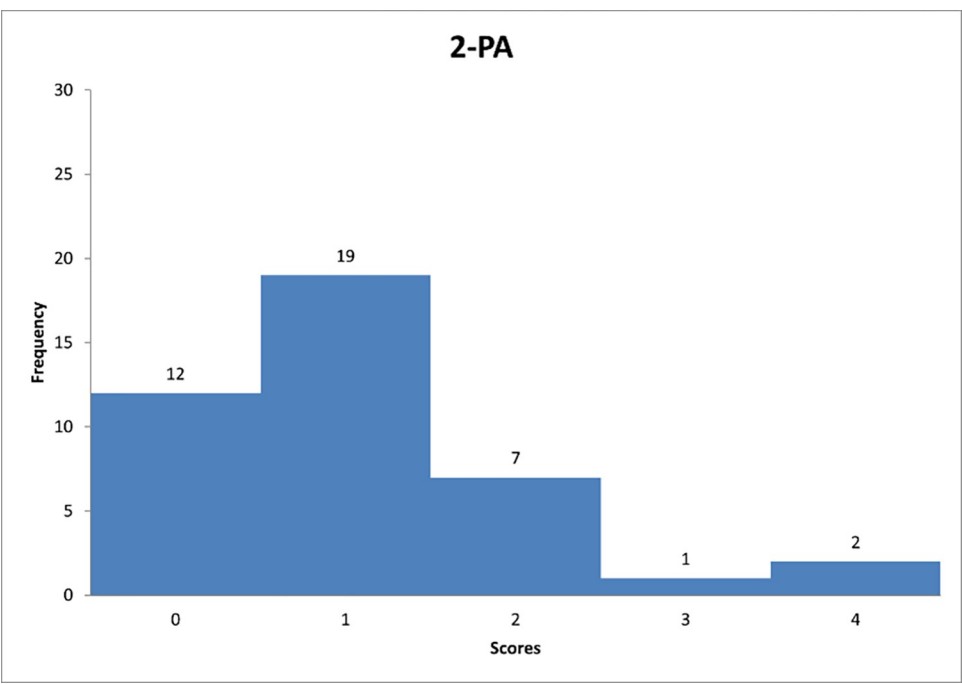

**Fig 14. Histograms based on respondents' propensity to adapt passive/avoidant style in relation to question PA-2.**

interest and the welfare of the group (59% of respondents). The average score for question 1 was 2,3. It can be concluded that the respondents showed a tendency to play sometimes the "trust-building" role included in the transformational leadership style. Fig 8 shows that the largest number of respondents showed a tendency to have "quite often" a sense of strength and

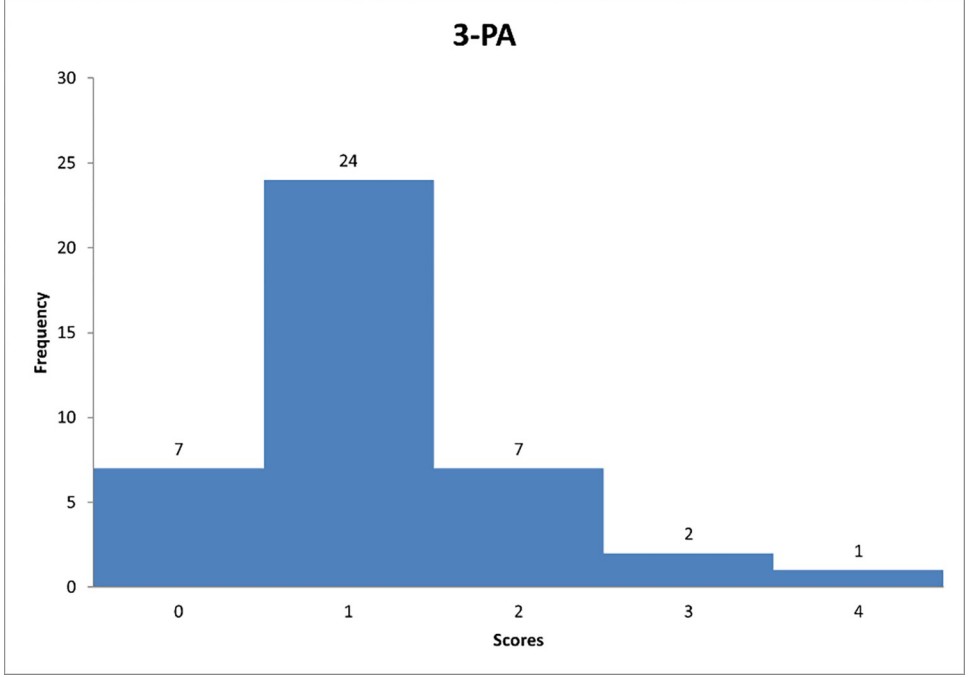

**Fig 15. Histograms based on respondents' propensity to adapt passive/avoidant style in relation to question PA-3.**

self-confidence. None of the respondents marked the answer "never", which can suggest that each person to some extent represents the researched trait of the leader. In Fig 9, the largest number of respondents allows subordinates to see problems from different perspectives also "quite often". Based on the calculated average score, it can be concluded that the respondents showed a tendency to display the feature "encourages innovative thinking" quite often, classified as the style of the transformational leader.

Next, Fig 10 shows that the largest number of respondents are managers who "quite often" clearly define goals and rewards related to their achievements. In Fig 11, it can be noted that the majority of respondents are for focusing on monitoring the mistakes of their subordinates. Fig 12 indicates that respondents again "quite often" follow the mistakes of their subordinates. The results indicate that the transactional leadership model is common on construction sites among construction managers or site managers.

Further analysis reveals that the most of respondents "sometimes" interfere with the problems of their subordinates when the so-called fighting fires (Fig 13). Fig 14 shows that decision-makers on construction sites "rarely" avoid making decisions. This group includes 46% of the respondents, which is almost half of the answers. In the last chart (Fig 15) one can observe that it is "rarely" deliberately delayed a decision-making process.

The following Tables 2–7 include more details. Most of the average scores are repeated and duplicate the research results for the entire sample.

**Table 2. Scores for particular styles (total answers).**

| Question-Style | 1-TF | 2-TF | 3-TF | 1-TA | 2-TA | 3-TA | 1-PA | 2-PA | 3-PA |
|---|---|---|---|---|---|---|---|---|---|
| Mean score | 2,3 | 3,0 | 2,9 | 3,1 | 2,6 | 2,7 | 2,0 | 1,1 | 1,2 |
| Average score | 2,7 | | | 2,8 | | | 1,4 | | |

**Table 3. Scores for particular styles (small companies).**

| Question-Style | 1-TF | 2-TF | 3-TF | 1-TA | 2-TA | 3-TA | 1-PA | 2-PA | 3-PA |
|---|---|---|---|---|---|---|---|---|---|
| Mean score | 2,6 | 2,6 | 2,6 | 3,0 | 2,7 | 2,9 | 1,9 | 1,6 | 1,4 |
| Average score | 2,6 | | | 2,9 | | | 1,6 | | |

**Table 4. Scores for particular styles (medium companies).**

| Question-Style | 1-TF | 2-TF | 3-TF | 1-TA | 2-TA | 3-TA | 1-PA | 2-PA | 3-PA |
|---|---|---|---|---|---|---|---|---|---|
| Mean score | 2,2 | 3,0 | 3,1 | 3,2 | 2,5 | 2,6 | 1,9 | 1,0 | 1,1 |
| Average score | 2,8 | | | 2,8 | | | 1,3 | | |

**Table 5. Scores for particular styles (large companies).**

| Question-Style | 1-TF | 2-TF | 3-TF | 1-TA | 2-TA | 3-TA | 1-PA | 2-PA | 3-PA |
|---|---|---|---|---|---|---|---|---|---|
| Mean score | 2,3 | 3,2 | 2,9 | 3,0 | 2,6 | 2,8 | 2,0 | 1,0 | 1,2 |
| Average score | 2,8 | | | 2,8 | | | 1,4 | | |

**Table 6. Scores for particular styles (females).**

| Question-Style | 1-TF | 2-TF | 3-TF | 1-TA | 2-TA | 3-TA | 1-PA | 2-PA | 3-PA |
|---|---|---|---|---|---|---|---|---|---|
| Mean score | 2,3 | 2,9 | 2,6 | 3,0 | 2,5 | 2,9 | 2,4 | 1,5 | 1,6 |
| Average score | 2,6 | | | 2,8 | | | 1,8 | | |

**Table 7. Scores for particular styles (males).**

| Question-Style | 1-TF | 2-TF | 3-TF | 1-TA | 2-TA | 3-TA | 1-PA | 2-PA | 3-PA |
|---|---|---|---|---|---|---|---|---|---|
| Mean score | 2,2 | 2,9 | 2,9 | 3,0 | 2,5 | 2,7 | 1,8 | 1,0 | 1,1 |
| Average score | 2,7 | | | 2,7 | | | 1,3 | | |

Only a certain noticeable change is brought by comparing the answers of the most and the least experienced construction managers. While the results for the PA and TA models are quite similar (Fig 16), in the case of the TF model, more experienced employees tend to almost be certain that they represent the characteristics of a transformational leader.

According to the collected surveys, it can be concluded that the respondents—construction managers or site managers show a mix of leadership styles. Neither of them exhibits only one, pure style—it is not possible.

## Discussion

### Current leadership models

The leading leadership models used on construction sites during the investment are mainly transactional and transformational. However, the features that are not related to the effective functioning of the investment are those belonging to the passive model. Therefore, the features of this style are rarely used by managers, as shown in the analysis of the conducted research. Due to their focus on effective execution, leaders should be driven primarily by a transactional or transformational style (depending on the situation) [71]. These models mean that a manager who properly combines their individual features will be efficient, leading an effective and successful group. When the situation requires it, he will be able to focus on the task and lead investments to meet the expected assumptions, but also be able to lead the group, focus on mutual relations and relationships. It will properly adjust to the staff to motivate and mobilize them as well as teach them to decide and to be willing to have more and more responsibility. Managers should make as little or no use of the features of the passive model as possible. This

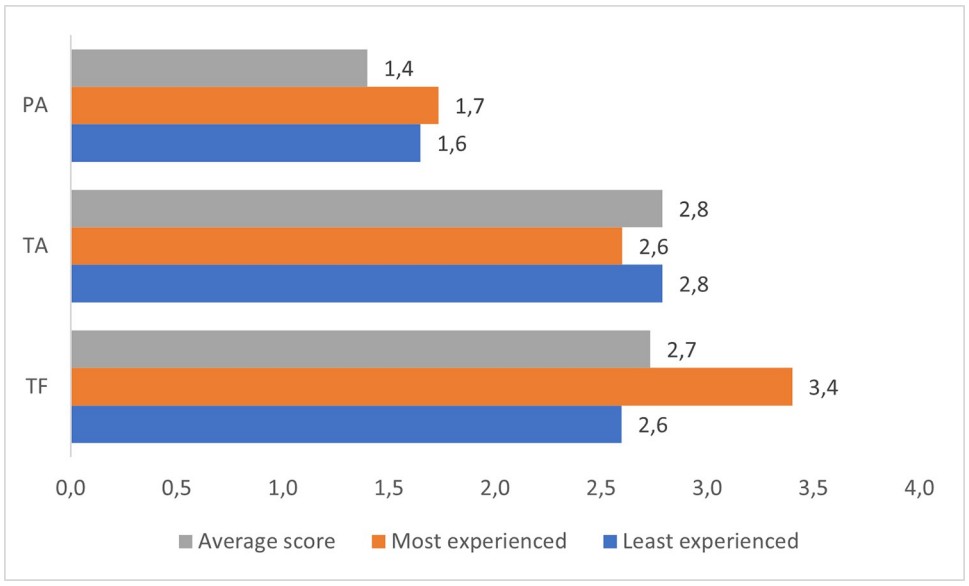

**Fig 16. Distribution of experience in managing construction works represented by the respondents.**

model is ineffective and subordinates have a negative attitude towards the leader with such inclinations.

The topic of the most popular leadership models is an interesting and still unexplored gap in improving the functioning of enterprises. That is why a lot of research has been carried out in the world and results that represent the dominant styles have been compiled. They concern the area of management in the range of each of the areas, not only in construction.

The presented results are consistent with the conclusions presented in the literature [72].

## Technology-oriented leadership models

**General background.** It should be noted that, on the one hand, technology is an opportunity that leads to increasing the efficiency of processes. On the other hand, however, it is a threat, it will be a surveillance tool, or a technique used for unfair competition, etc. Therefore, how to deal with this dilemma? First, one should focus on education as a dimension of implementing strategies to deal with the threats arising from the development of technology. This article is devoted to searching for a way to shape the technological maturity of a construction company expressed by the harmonized state of competence of all employees in the field of technology used in the enterprise. Moreover, a role of leadership as an enabler for implementing new technologies was identified [48].

**Basic attributes of models.** Fig 17 presents a case where there is a lack of harmony in the values represented by the employees of a given enterprise. A change leader is emerging—the so-called leader of the technology development in a construction company, who can be called construction technology manager (CTM). His/her competences are indisputable, he/she can be a graduate of post-graduate studies in the field of IT for construction, construction technology or a participant of training/workshops on such topics. At this time, only he/she and the top management (because they hired him) are aware of the advantages of managing computer-aided construction projects. The skills and knowledge of the leader are usually divergent with the skills and knowledge of other employees. The construction technology manager's

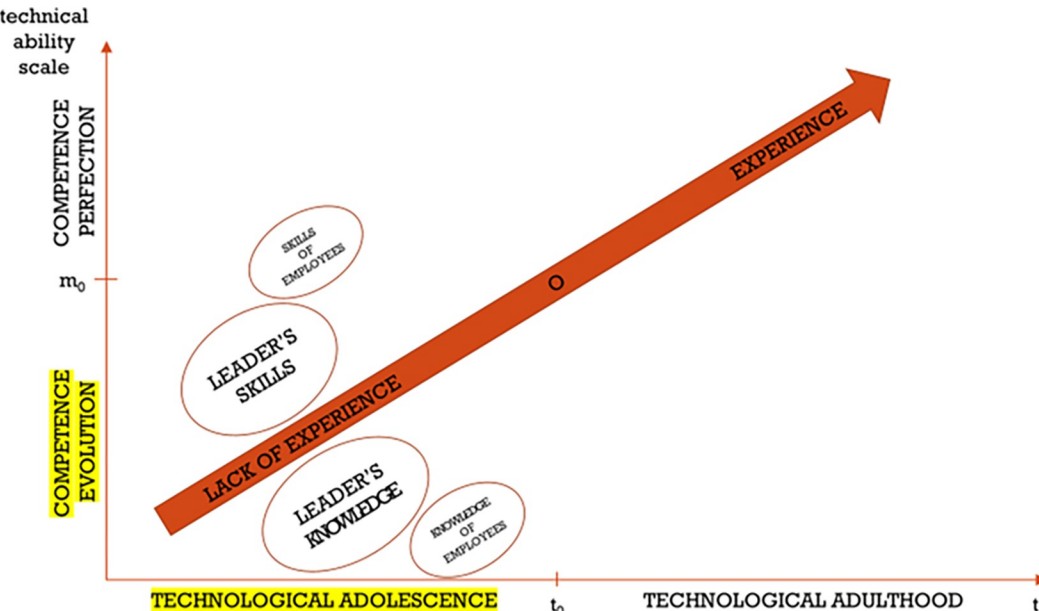

**Fig 17. The pursuit of high perfection in meeting technological requirements in construction companies—Stage one: Lack of experience.**

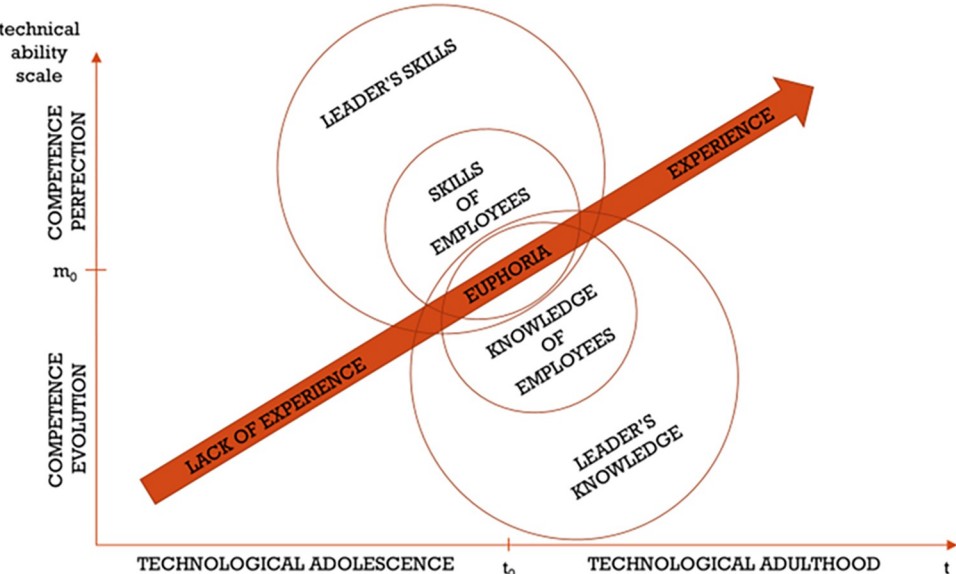

**Fig 18. The pursuit of high perfection in meeting technological requirements in construction companies—Stage two: Euphoria.**

effort should be directed towards sharing these qualities with others and educating them. This stage is characterized by the evolution of technological maturity, and when it comes to the period it can be called technological adolescence of a company. The success of this phase depends on the level of commitment of the leader, the belief in the correctness of matters, as well as the consistent pursuit of change.

The next stage is the euphoria (Fig 18). After a monotonous period of internal organizational changes, employee training, discussions, consultations, reorganization, and sometimes reengineering in a construction company, the phase is reached where all skills and knowledge of the leader and other employees become shared. The construction technology manager's effort should be directed towards maintaining a state of euphoria. This moment $M(t_0)$ can be called technological-maturity-equilibrium. Further actions must lead to the perfection of actions and, associated with it, the full experience level.

The full experience level (Fig 19) that can be observed in construction companies, related to achieving perfection in mastering competence in terms of technology by all employees, is the final stage of the process. It crowns the efforts of CTM, as well as all employees who have gained the conviction that their skills and knowledge are common to all. The newly created corporate culture can be a condition for the success of a construction company that uses the advantages of competence perfection in technology connected with construction.

However, for the progress described above to be realistic, it must be conditioned by appropriate leadership models.

## Modelling evolutionary leadership

Effective business activities depend on the quality of individual employee behaviour [73]. To effectively manage construction processes, it is necessary to apply appropriate means to use power and impact the subordinates in an efficient way. Being a leader is not an easy task [74]. There are many leadership theories with various leadership styles (autocratic, democratic, laissez-faire). In the organizational and project domains, a leadership competency is considered an ability to convert knowledge into action that results in a desired performance [75].

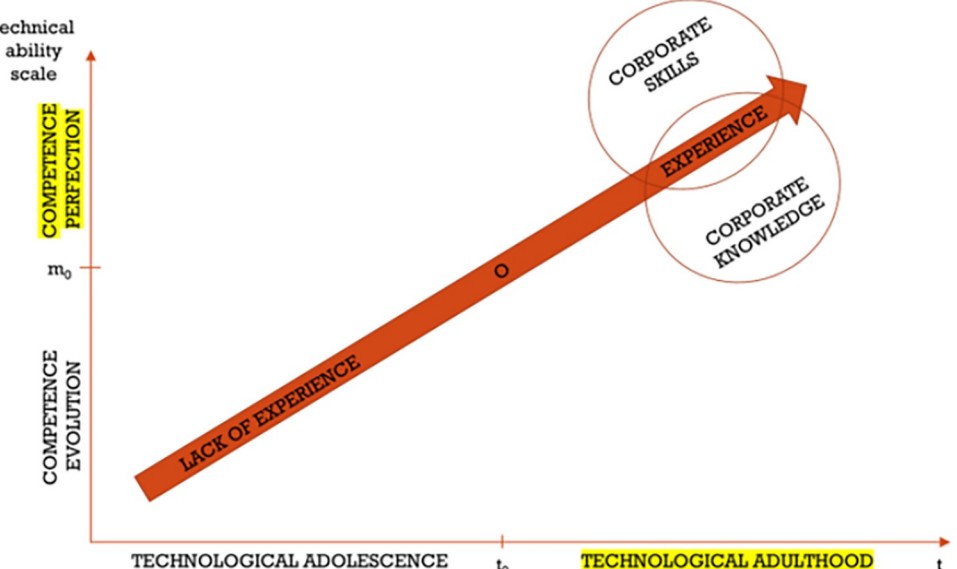

**Fig 19. The pursuit of high perfection in meeting technological requirements in construction companies—Stage three: Full experience.**

A choice of the right strategy should be guided by the achievements of modern science, including evolutionary leadership theory [76]. The attitude, developed by van Vugt and Ahuja, is based on the following assumption: both leadership and followership are important for the reproductive success of ancestors. However, modern organizational structures of companies of all industries are sometimes inconsistent with innate psychological mechanisms of two processes: leading and following. This explains some problems in the relationship between managers and subordinates. The leadership is conditioned by the fact that turning subordinates (employees) into followers of the leader is a key success factor for organizations.

An evolutionary leadership model, configured for technology knowledge management in a construction company, is shown in Fig 20. It can be noted that an appropriate approach at the

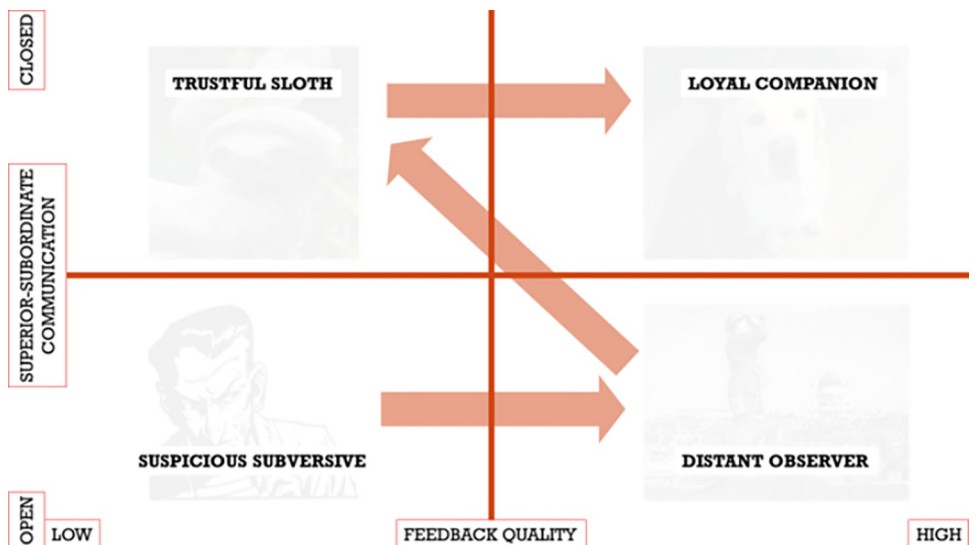

**Fig 20. Evolutionary leadership model for effective knowledge diffusion about technology in a construction company.**

beginning is to use open communication model for superior-subordinate. Then a feedback quality increases and the behaviour of the employee switches into a distant observer.

Once all the communication model converts into a closer one, once again a feedback quality decreases as the subordinates become more suspicious again. When everybody in the organization starts to understand how to use all advantages of the technology in a construction company, the feedback quality can increase one more time and an employee becomes a loyal companion as he/she trusts in the leader and his/her position.

Accordingly investigated survey data shows that "transactional" and "transformational" leadership types dominate over "passive/avoidant" patterns, however, what was revealed, there is no single best 'one-fits-all' leadership approach that could be used within a random construction project organization. On the other hand, this is in line with the evolutionary leadership model [77], being an effective knowledge diffusion tool and an enabler for improving the maturity [78] of technological issues in a construction company, what was proposed in this study. In this respect, it is thought that the dissemination of the study results in the construction sector will contribute to the knowledge at a global scale.

## Conclusions

Transforming a construction company into an organization that cares about technology development and diffusion of the related competencies is not easy. It requires time, experience and persistence in reaching established targets. Going towards technological adulthood is not only about conducting a series of training sessions about BIM, Big Data, drones etc. with certificates. It is not just about creating new procedures. This transformation is a process of getting experience in which the skills and knowledge of the whole corporation are the same as what the employee can and knows.

The representatives of construction projects make it clear that they do not prefer one particular model of leadership. By confronting the conclusions of the study with the literature, one can see a regularity regarding the dominance of the transformational-transactional style. On the other hand, the digital revolution experienced by the construction industry requires completely new management tools that can be discussed in subsequent editions of scientific research.

The evolutionary leadership model seems to be the most rational of all. It is obvious that the leadership may be a subject of criticism. Leaders may be the wrong example. They can trick and lie. But in the end, one should trust them to achieve some measurable benefits. The basic concept of evolutionary leadership including four original states: suspicious subversive, distant observer, trustful sloth and loyal companion was adapted to new circumstances of effective knowledge diffusion of technology development in a construction company.

## Supporting information

**S1 Data. Source data.**
(PDF)

## Author Contributions

**Conceptualization:** Jarosław Górecki, Jadwiga Bizon-Górecka.

**Data curation:** Jarosław Górecki.

**Formal analysis:** Jarosław Górecki, Jadwiga Bizon-Górecka, Abdullah Emre Keleş.

**Funding acquisition:** Jarosław Górecki.

**Investigation:** Jarosław Górecki, Abdullah Emre Keleş.

**Methodology:** Jarosław Górecki.

**Project administration:** Jarosław Górecki.

**Resources:** Jarosław Górecki, Ewa Bojarowicz.

**Software:** Jarosław Górecki.

**Supervision:** Jarosław Górecki.

**Validation:** Jarosław Górecki, Ewa Bojarowicz, Jadwiga Bizon-Górecka, Umer Zaman.

**Visualization:** Jarosław Górecki, Abdullah Emre Keleş.

**Writing – original draft:** Jarosław Górecki, Jadwiga Bizon-Górecka.

**Writing – review & editing:** Jarosław Górecki, Umer Zaman, Abdullah Emre Keleş.

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
