## [Decision Letter · Decision Letter 0]

28 Mar 2022

PONE-D-21-35766Leadership models in era of new technological challenges in construction projectsPLOS ONE

Dear Dr. Górecki,

Thank you for submitting your manuscript to PLOS ONE. After careful consideration, we feel that it has merit but does not fully meet PLOS ONE’s publication criteria as it currently stands. Therefore, we invite you to submit a revised version of the manuscript that addresses the points raised during the review process.

We look forward to receiving your revised manuscript.

Kind regards,

Xingwei Li, Ph.D.

Academic Editor

PLOS ONE

Journal Requirements:

We thank the UTP University of Science and Technology, Bydgoszcz (Poland) and its Faculty of Civil and Environmental Engineering and Architecture for proving funds for publication of the manuscript.

The authors received no specific funding for this work.

Additional Editor Comments:

I agree with the reviewers and recommend a major revision of this manuscript.

Reviewers' comments:

Reviewer #1: The paper has a great new idea of predicting the path for future Leadership shape. The predictions are not supported in the study.

Also if the predictions are supported what is the guidelines by which we can help construction leaders to understand these trends of preditions.

Reviewer #2: The research addresses an interesting and worthy topic. The angle of the research is also timely as leaders do play a key role in furthering IT in construction industry. I also think the survey and data are good enough to support the research purpose.

A major concern, however, is about the writing and organization of Section 1 and 2. Although the references are relevant, the way it is organized and written is so casual and fragmented. I couldn't see how the arguments are built to substantiate the research.

Abstract: Out of focus.

Keywords: leadership should be one of the keywords.

1. Introduction

It is loosely written - all the terms, LCCA, HVAC, AI, 3D, VR, AR, and et al were used but not connected.

2. Technology challenges in the construction project:

Is this the appropriate heading for this research? The context is also very disconected.

3 Materials and Methods:

(1) I am not sure whether the paragraph between the line 171-184 is an interpretation of the cited paper from [36] and the same for the following paragraph between the lines 185 and 191 [37].

(2) I would expect the paragraphs between the lines 158 and 200 to describe how the questions are designed. However the way it is written is simply an extension of relevant literature by only using two references [36] and [37]

(3) How were the people identified? How are the questions designed?

6. Conclusions

The conclusions are disconnected from the research findings.

Reviewer #3: The authors brought up an important question that how different kinds of leadership shape the digital transformation in construction industry. The beginning and ending part of this paper is well written. However, the methodology, questionnaire development and data analysis seemed to be separated parts from the rest of the paper. For example, how simple analysis of distributions and averages will illustrate the influence of leadership types? Is there a significance analysis to argue for statistically significant results? Will the size of companies has an impact on leadership types and technology developments? Hope the authors could explain further in the future.

---

## [Author Response · Author response to Decision Letter 0]

21 Sep 2022

RESPONSE TO REVIEWER #1

Dear Reviewer,

As the authors of the original paper entitled “Leadership models in era of new technological challenges in construction projects”, submitted to PlosOne, we would like to thank you for the feedback on our research.

COMMENTS:

The paper has a great new idea of predicting the path for future Leadership shape. The predictions are not supported in the study.

Also if the predictions are supported what is the guidelines by which we can help construction leaders to understand these trends of preditions.

REPLY:

Your comments motivated us to fully rebuild our manuscript. We hope you are satisfied with the updated version.

- Authors

RESPONSE TO REVIEWER #2

Dear Reviewer,

As the authors of the original paper entitled “Leadership models in era of new technological challenges in construction projects”, submitted to PlosOne, we would like to thank you for the feedback on our research.

COMMENTS:

The research addresses an interesting and worthy topic. The angle of the research is also timely as leaders do play a key role in furthering IT in construction industry. I also think the survey and data are good enough to support the research purpose.

A major concern, however, is about the writing and organization of Section 1 and 2. Although the references are relevant, the way it is organized and written is so casual and fragmented. I couldn't see how the arguments are built to substantiate the research.

Abstract: Out of focus.

Keywords: leadership should be one of the keywords.

1. Introduction

It is loosely written - all the terms, LCCA, HVAC, AI, 3D, VR, AR, and et al were used but not connected.

2. Technology challenges in the construction project:

Is this the appropriate heading for this research? The context is also very disconected.

3 Materials and Methods:

(1) I am not sure whether the paragraph between the line 171-184 is an interpretation of the cited paper from [36] and the same for the following paragraph between the lines 185 and 191 [37].

(2) I would expect the paragraphs between the lines 158 and 200 to describe how the questions are designed. However the way it is written is simply an extension of relevant literature by only using two references [36] and [37]

(3) How were the people identified? How are the questions designed?

6. Conclusions

The conclusions are disconnected from the research findings.

REPLY:

Your comments motivated us to fully rebuild our manuscript. 

We added several crucial references. 

We hope you are satisfied with the updated version.

- The abstract and keywords were changed.

- The introduction was modified to underline what we concentrate on.

- The second chapter was changed.

- The third chapter was modified according to your suggestions.

- We hope the modified Conclusions are within the research findings.

- Authors

RESPONSE TO REVIEWER #3

Dear Reviewer,

As the authors of the original paper entitled “Leadership models in era of new technological challenges in construction projects”, submitted to PlosOne, we would like to thank you for the feedback on our research.

COMMENTS:

The authors brought up an important question that how different kinds of leadership shape the digital transformation in construction industry. The beginning and ending part of this paper is well written. However, the methodology, questionnaire development and data analysis seemed to be separated parts from the rest of the paper. For example, how simple analysis of distributions and averages will illustrate the influence of leadership types? Is there a significance analysis to argue for statistically significant results? Will the size of companies has an impact on leadership types and technology developments? Hope the authors could explain further in the future.

REPLY:

Your comments motivated us to fully rebuild our manuscript. We added several crucial references to eliminate the gaps. 

We hope you are satisfied with the updated version.

- The abstract and keywords were changed.

- The introduction was modified to underline what we concentrate on.

- The second chapter was changed.

- The third chapter was modified according to your suggestions.

- We hope the modified Conclusions are within the research findings.

We hope our explanations make the manuscript more consistent.

- Authors

---

## [Decision Letter · Decision Letter 1]

1 Nov 2022

PONE-D-21-35766R1Leadership models in era of new technological challenges in construction projectsPLOS ONE

Dear Dr. Górecki,

Thank you for submitting your manuscript to PLOS ONE. After careful consideration, we feel that it has merit but does not fully meet PLOS ONE’s publication criteria as it currently stands. Therefore, we invite you to submit a revised version of the manuscript that addresses the points raised during the review process.

We look forward to receiving your revised manuscript.

Kind regards,

Xingwei Li, Ph.D.

Academic Editor

PLOS ONE

Journal Requirements:

Additional Editor Comments:

Ask the author to explain the scale and survey results in more detail.

Reviewers' comments:

Reviewer's Responses to Questions

**Comments to the Author**

1. If the authors have adequately addressed your comments raised in a previous round of review and you feel that this manuscript is now acceptable for publication, you may indicate that here to bypass the “Comments to the Author” section, enter your conflict of interest statement in the “Confidential to Editor” section, and submit your "Accept" recommendation.

Reviewer #1: All comments have been addressed

Reviewer #3: (No Response)

2. Is the manuscript technically sound, and do the data support the conclusions?

Reviewer #1: Partly

Reviewer #3: Partly

3. Has the statistical analysis been performed appropriately and rigorously? 

Reviewer #1: Yes

Reviewer #3: Yes

4. Have the authors made all data underlying the findings in their manuscript fully available?

Reviewer #1: Yes

Reviewer #3: Yes

5. Is the manuscript presented in an intelligible fashion and written in standard English?

Reviewer #1: Yes

Reviewer #3: Yes

6. Review Comments to the Author

Reviewer #1: I would like to comment at the paper as there are many versions of the paper in the submittal. I would be better to have the latest version in the submittal.

Reviewer #3: The authors have addressed most of the reviewers' concerns. However, they should take the following questions into consideration:

1. They need to explain how the 9 questions reflect the three kinds of leadership models. Is there existing research that validated the scale? Are the three questions enough to describe the leadership models?

2. There is lack of explanation of how the questionnaire results are related to the leadership model proposed in Section 5.1.

7. PLOS authors have the option to publish the peer review history of their article (what does this mean?). If published, this will include your full peer review and any attached files.

Reviewer #1: No

Reviewer #3: No

---

## [Author Response · Author response to Decision Letter 1]

24 Nov 2022

RESPONSE TO REVIEWER #1

COMMENTS:

I would like to comment at the paper as there are many versions of the paper in the submittal. I would be better to have the latest version in the submittal.

REPLY:

We did our best that the initial submission as well as the resubmission had followed the journal's requirements. We believe that this time the revised version of the paper will be properly presented. It seems that authors don't have much impact on how files are presented in the submission system. We hope the final version of the paper will satisfy the reviewers.

RESPONSE TO REVIEWER #2

COMMENTS:

The authors have addressed most of the reviewers' concerns. However, they should take the following questions into consideration:

1. They need to explain how the 9 questions reflect the three kinds of leadership models. Is there existing research that validated the scale? Are the three questions enough to describe the leadership models?

2. There is lack of explanation of how the questionnaire results are related to the leadership model proposed in Section 5.1.

REPLY:

The multiple leadership questionnaire (MLQ) scale developed by Bass and Avolio and used by one of the authors in his paper [Keleş et al, 2021] was used as the basis for determining the questionnaire questions of this study [Bass and Avolio, 1995]. The questionnaire defines the 3 leadership types discussed in this study and consists of 45 statements on a 5-point Likert scale. In the research, 9 of 45 questions, which were the most decisive, were selected and asked the participants during a session of the questionnaire. In this respect, it is clear that the answers to these questions can be used in determining the leadership types aimed at the study. This information has been added to the "Materials and Methods" section (lines 295-301).

As a result of the survey, it is seen that "transactional" and "transformational" leadership types dominate over “passive/avoidant” patterns, however, what was revealed, there is no single best ‘one-fits-all’ leadership approach that could be used within a random construction project organization. On the other hand, this is in line with the evolutionary leadership model [Hersey and Blanchard, 1969], being an effective knowledge diffusion tool and an enabler for improving the maturity [Serna, 2015] of technological issues in a construction company, what was proposed in this study. In this respect, it is thought that the dissemination of the study results in the construction sector will contribute to the knowledge at a global scale. Such an explanation was added in lines 536-543.

Thank you for your valuable contribution. We believe our manuscript has been enriched even more.

- Authors

---

## [Editor Report · Decision Letter 2]

28 Nov 2022

Leadership models in era of new technological challenges in construction projects

PONE-D-21-35766R2

Dear Dr. Górecki,

We’re pleased to inform you that your manuscript has been judged scientifically suitable for publication and will be formally accepted for publication once it meets all outstanding technical requirements.

Kind regards,

Xingwei Li, Ph.D.

Academic Editor

PLOS ONE
---

## [Editor Report · Acceptance letter]

6 Dec 2022

PONE-D-21-35766R2 

Leadership models in era of new technological challenges in construction projects 

Dear Dr. Górecki:

I'm pleased to inform you that your manuscript has been deemed suitable for publication in PLOS ONE. Congratulations! Your manuscript is now with our production department. 

Kind regards, 

on behalf of

Prof. Dr. Xingwei Li 

Academic Editor

PLOS ONE